# Nano-Bioremediation of Arsenic and Its Effect on the Biological Activity and Growth of Maize Plants Grown in Highly Arsenic-Contaminated Soil

**DOI:** 10.3390/nano14131164

**Published:** 2024-07-08

**Authors:** Mahmoud El Sharkawy, Arwa A. AL-Huqail, Alya M. Aljuaid, Nourhan Kamal, Esawy Mahmoud, Alaa El-Dein Omara, Nasser Abd El-Kader, Jian Li, Nashaat N. Mahmoud, Ahmed A. El Baroudy, Adel M. Ghoneim, Sahar Mohamed Ismail

**Affiliations:** 1School of the Environment and Safety Engineering, School of Emergency Management, Jiangsu University, Zhenjiang 212013, Chinajianli@ujs.edu.cn (J.L.); 2Soil and Water Department, Faculty of Agriculture, Tanta University, Tanta 31511, Egypt; nourhan.6694@agr.tanta.edu.eg (N.K.); ahmed.elbaroudy@agr.tanta.edu.eg (A.A.E.B.); 3Department of Biology, College of Science, Princess Nourah bint Abdulrahman University, P.O. Box 84428, Riyadh 11671, Saudi Arabia; aaalhuqail@pnu.edu.sa; 4Biology Department, College of Science and Humanities, Shaqra University, Shaqra 15571, Saudi Arabia; 5Agricultural Research Center, Department of Microbiology, Soils, Water and Environment Research Institute, Giza 12112, Egypt; alaa.ahmed@arc.sci.eg; 6Department of Botany and Microbiology, Faculty of Science, Al-Azhar University, Nasr City, Cairo 11884, Egypt; drmahmoud_123@yahoo.com; 7Agricultural Research Center, Field Crops Research Institute, Cairo 12619, Egypt; 8Soil Physics and Chemistry Department, Desert Research Center, Cairo 11753, Egypt; ziadmoaz1122@gmail.com

**Keywords:** nanobioremediation, arsenic, biological activity, as-tolerant bacterial isolates, maize

## Abstract

Arsenic (As)-contaminated soil reduces soil quality and leads to soil degradation, and traditional remediation strategies are expensive or typically produce hazardous by-products that have negative impacts on ecosystems. Therefore, this investigation attempts to assess the impact of As-tolerant bacterial isolates via a bacterial *Rhizobim nepotum* strain (B_1_), a bacterial *Glutamicibacter halophytocola strain* (B_2_), and MgO-NPs (N) and their combinations on the arsenic content, biological activity, and growth characteristics of maize plants cultivated in highly As-contaminated soil (300 mg As Kg^−1^). The results indicated that the spectroscopic characterization of MgO-NPs contained functional groups (e.g., Mg-O, OH, and Si-O-Si) and possessed a large surface area. Under As stress, its addition boosted the growth of plants, biomass, and chlorophyll levels while decreasing As uptake. Co-inoculation of *R. nepotum* and *G. halophytocola* had the highest significant values for chlorophyll content, soil organic matter (SOM), microbial biomass (MBC), dehydrogenase activity (DHA), and total number of bacteria compared to other treatments, which played an essential role in increasing maize growth. The addition of *R. nepotum* and *G. halophytocola* alone or in combination with MgO-NPs significantly decreased As uptake and increased the biological activity and growth characteristics of maize plants cultivated in highly arsenic-contaminated soil. Considering the results of this investigation, the combination of *G. halophytocola* with MgO-NPs can be used as a nanobioremediation strategy for remediating severely arsenic-contaminated soil and also improving the biological activity and growth parameters of maize plants.

## 1. Introduction

Arsenic (As) is a poisonous element that poses a serious threat to the ecosystem and the health of humans globally. It is particularly harmful when present in inorganic forms in soil [1]. Arsenic has become a concern for the public’s health and the environment as a result of soil or water contamination. This is because arsenic is extremely poisonous, does not biodegrade, can accumulate in the body, and may cause cancer. The International Agency for Research on Cancer has categorized inorganic arsenic as a Class I carcinogen [2]. High As levels in soil can affect its characteristics and fertility, kill helpful bacteria, and disrupt the biology of plants, preventing growth and development. This has an impact on the food chain and the entire ecosystem [3]. The World Health Organization has determined that arsenic concentrations in plants grown in soils without arsenic range from 0.02 to 5 mg kg^−^^1^ (dry weight). The daily intake of total arsenic from foods and beverages is generally in the range of 20–300 μg/day. Soil contamination with arsenic can come from natural sources such as geochemical processes of rocks and minerals or volcanic emissions and biological processes. There are other sources resulting from human activity, including metal smelting, mining, pesticides, irrigation from sewage, and untreated industrial water from farmers such as in the Mahalla El-Kobra area, Egypt [4,5].

Traditional soil remediation strategies for As such as removal excavation and landfilling and containment (capping) are either high in the cost of removal or the loss of land use, and therefore, the use of nanotechnology and biological-based techniques for on-site arsenic removal is desirable due to cost and sustainability issues that have not yet been studied together [6]. Plant growth-promoting rhizobacteria (PGPR) are considered beneficial soil microorganisms that have the potential to improve plant growth and increase resistance to both biotic and abiotic stresses. Microorganisms are widely known for producing a variety of polysaccharides, including internal polysaccharides, functional polysaccharides, and extracellular or exopolysaccharides (EPSs). Microorganisms create EPSs as a defense biomacromolecule in response to strong stress situations such as food scarcity, salt tolerance, heat stress, drought, and heavy metal pollution [7,8]. Particularly, plant microbiota can encourage growth, trigger a defense mechanism against pests and diseases, and improve resistance to abiotic stresses like heavy metals, salinity (EC), and drought [9,10].

Many soil-based microorganisms in As-contaminated areas have developed adaptation mechanisms to survive in a polluted environment, most notably by utilizing toxic arsenic as a resource in metabolism and reproduction. Some bacteria, such as *Agrobacterium*, *Burkholderia*, *Bacillus*, and *Rhizobium*, are appropriate options for the bioremediation of As and plant growth-promoting rhizobacterium (PGPR) [11,12].

Some rhizobacteria are characterized by phytohormone synthesis, siderophore production, and the fixation of nitrogen, soluble phosphorus, and other nutrients, which alter the transport and availability of metals (loids) in the soil [13] and PGPR [14]. Inoculation of PGPR with poplar may reduce arsenic toxicity and increase the efficiency of phytoremediation in arsenic-contaminated soils. *Glutamicibacter halophytocola* strain KLBMP 5180 may serve a significant beneficial ecological purpose via improved growth and salt resistance in host plants using physiological and molecular mechanisms [15]. Qaralleh et al. [16] found that *R. nepotum* is an efficient phenol degrader that tolerates doses of up to 1200 ppm. Plants inoculated with *R. nepotum* strain ACO-5A showed significantly enhanced growth of century plants (Agave americana L.) and increased soil nutrient availability [17]. Herliana et al. [18] found a significant increase in leaf area, pod weight, and number of pods in black soybean (*Glycine max* (L.)) when inoculated with *R. nepotum*.

Nano-bioremediation is an innovative approach that combines nanotechnology with bioremediation to achieve a more beneficial, cost-effective, and environmentally conscious restoration than either procedure alone. The nanoparticles have a great ability to remove these dangerous pollutants and provide a nutrient-rich substrate that promotes microbiological activity, raising the standard regarding environmental remediation [19]. The elimination of harmful contaminants by nanobioremediation is effective due to some of the wonderful qualities of nanoparticles, which include the large surface area, high adsorption capacity, and environmental friendliness. These particular qualities play a crucial role in cleaning up pollutants from the environment. Nanoparticles such as Fe, ZnO, TiO2, and Ag are very useful for cleaning up very hazardous contaminants, including DDT, carbamates, As, Cd, Cr, and Pb, from the soil [20]. Němeček et al. [21] found that the combination of nanoscale zero-valent iron (nZVI) and whey reduces Cr(VI) to approximately 97–99%. Adikesavan and Nilanjana [22] found an increased degradation rate and high efficiency in treating cefdinir (a semisynthetic cephalosporin antibiotic) in aqueous media using MgO nanoparticles with yeast. Therefore, the novelty of this research is the use *of Rhizobim nepotum*, *Glutamicibacter halophytocola*, MgO-NPs, and their combinations in highly arsenic-contaminated soil to remediate arsenic content and increase maize growth. The present study assumed that the addition of MgO-NPs alone would not be able to completely reduce arsenic content in highly arsenic-contaminated soil and increase maize growth. Therefore, the aim of the study was to investigate the impact of As-tolerant bacterial isolates via a bacterial *R. nepotum* strain (B_1_), a bacterial *G. halophytocola* strain (B_2_), nano-MgO (N), and their combinations on the As content, biological activity, and growth parameters of maize plants grown in highly As-contaminated soil (300 mg As Kg^−^^1^).

## 2. Materials and Methods

### 2.1. Soil Used in this Experimental

The soil that was utilized for the present study was taken from the top 30 cm layer of newly reclaimed soil in El-Behira Governorate, Egypt. The area is located at 30°47′09″ N and 31°00′02″ E. Table 1 describes the physical and chemical properties of the sandy soil.

### 2.2. Bacteria Isolation and Purification

Using the enrichment culture approach, As-resistant bacteria were isolated from the obtained soil samples on Nutrient Agar (NA) plates that had been enriched with 300 mg L^−^^1^ of arsenic from sodium arsenic (AsHNa_2_O_4_·7H_2_O) [23]. The soil samples were serially diluted and then plated and incubated at 30 °C. The three-way streak plate approach was used to select and purify the colonies showing proper plate growth. After carefully choosing a single colony, it was further purified using the streak plate procedure and kept on NA slants in order to conduct further research. The isolates of bacteria were identified biochemically using a VITEK 2 compact apparatus for bacterial identification and genetically via 16S ribosomal RNA (16S rRNA) sequencing.

### 2.3. Bacterial Inoculums

Two strains of bacteria (B_1_: *Rhizobim nepotum* and B_2_: *Glutamicibacter halophytocola*) were used as soil inoculums, which were isolated from soil irrigated with polluted drainage water for a period of over 10 years at El-Mahla El-Kobra, El-Gharbia Governate, Egypt [5].

### 2.4. Molecular Identification of Two Isolates

Using the 16S rRNA gene sequence (1.5 kpb), we identified two isolates of bacteria that are most tolerant to high concentrations of arsenic of up to 300 mg L^−^^1^, which are *Rhizobim nepotum* (B_1_) and *Glutamicibacter halophytocola* (B_2_) (Figure 1).

### 2.5. Preparation of Nano-MgO 

Nano-MgO was prepared according to [24]. Two solutions were prepared, the first by dissolving 20 mM of MgCl_2_ in 50 mL of ethanol. The second was prepared by dissolving 40 mM of NaOH in 50 mL of ethanol. Both solutions were stirred at 60 °C for 30 min. During stirring for 10 min, the second solution was added dropwise to the first solution at 60 °C, diluting the solution by adding 20 mL of ethanol with continuous stirring. Then the previous solution was centrifuged at 2000 rpm for 10 min, and the resulting residue was dried at 60 °C in a hot air oven. Then the resulting compound was calcined at 300 °C for 1 h to obtain crystalline MgO nanoparticles.

### 2.6. Pot Experimental Procedures

The experiment was performed in pots, each with 3 kg of sandy soil. Then, the sandy soil was washed five times with distilled water and left to dry after being washed for 24 h. The seeds of maize cultivar var. Y.Sc. 168 (6 seeds per pot) were sown in the pots (15 cm high, 17 cm internal diameter) in various situations for experiments. The treatments were as follows: (1) A: arsenic 300 ppm; (2) AB_1_: arsenic 300 ppm + *R. nepotum*; (3) AB_2_: arsenic 300 ppm + *G. halophytocola*; (4) AN: arsenic 300 ppm + Nano-MgO; (5) AB_1_N: arsenic 300 ppm + *R. nepotum* + Nano-MgO; (6) AB_2_N: arsenic 300 ppm + *G. halophytocola* + Nano-MgO; and (7) AB_1_B_2_: arsenic 300 ppm + *R. nepotum* + *G. halophytocola*.

Arsenic concentrations (300 mg L^−^^1^ of arsenic from sodium arsenic (AsHNa_2_O_4_.7H_2_O)) were added after germination until the length of the plant reached 2 cm. Nano-MgO was added at 2 mg kg^−^^1^, and strains *R. nepotum* and *G. halophytocola* were added at 1 × 10^9^ CFU/mL by pipet to the roots of plants. Distilled water was utilized to irrigate every treated plant at field capacity because the top water at this location is groundwater, which contains a percentage of arsenic (the amount of soil moisture or water content held in the soil after excess water has drained off and the rate of downward movement decreased significantly, which usually occurs within 2–3 days after irrigation). Fertilization and other agricultural practices were performed according to the Egyptian Ministry of Agriculture recommendation for maize plants in the Middle Delta area. The length of the root and shoot, as well as the total chlorophyll, were determined at 40 days, and then the maize was harvested. Maize growth during the experimental period is shown in Figure 2.

### 2.7. Biochemical Determination

Soil organic carbon (SOC) based on the [25] chromic acid wet oxidation method was used to determine soil organic matter as described by [26] after wet digestion, according to [27]. Soil pH was measured in a 1:2.5 (soil–water) suspension using a pH meter (Thermo-Fisher (HANNA-H12211-02), Waltham, MA, USA). Available arsenic was extracted using a diethylenetriaminepentaacetic acid (DTPA) solution and measured using an inductively coupled plasma atomic emission spectrometer (ICP-ISO Prodigy Plus, Teledyne CETAC Technologies, Omaha, NE, USA) with calibration by a standard curve. A stock solution of arsenic at a concentration of 1.000 g L^−^^1^ was used to prepare the standards. The calibration was adjusted in the range of 0–10 μg L^−^^1^, according to [28].

The total quantity of bacteria was determined using the extract of soil on agar media and expressed as CFU (log 10) g^−^^1^ dry soil at 30 days following sowing, in accordance with previous [29] techniques. The method outlined by [30] was used to determine the dehydrogenase activity in the soil samples. 

Microbial biomass carbon (MBC) was determined by the Chloroform fumigation-extraction method (CFEM) as reported by Amato and Ladd [31]. The following equation was used to calculate MBC: MBC = (EC fumigated soil − EC un-fumigated soil)/Kc
where EC = Extractable carbon; Kc = 0.379 (Kc is K_2_SO_4_ extraction efficiency) [32].

### 2.8. Plant Analysis

Total chlorophyll was determined using a chlorophyll meter (SPAD-502 Plus, Konica Minolta Optics, Inc., Tokyo, Japan) a day before harvesting. Using Digital Weighing Scales, the dried weights of the shoots and roots were measured to calculate the biomass of the plant. Plant materials (shoot and root) were dried at 70 °C for 24 h. The plant As uptake (influx into plants) (mg As pot^−1^) was calculated as the dry weight of plants and the As concentration of plants (total As content). As uptake (mg As pot^−1^) = (DW × %As/100) × 1000. Weights of 0.1 g from plant samples were digested using H_2_SO_4_ and HClO_4_. Total arsenic was determined during sample digestion using an inductively coupled plasma atomic emission spectrometer (ICP-ISO Prodigy Plus) in digested samples, according to Cottenie et al. [28]. 

### 2.9. Spectroscopic Analysis

The size of the nano-MgO was assessed using transmission electron microscopy (TEM), which was performed at the Electron Microscope Unit at Mansoura University in Egypt using a microscope FEI TECNAI G20 (200KV-LaB6 emitter, Hillsboro, OR, USA). The surface morphology of the nano-MgO was investigated using the JEOL (JSM-7610F FEG-SEM, Tokyo, Japan) scanning electron microscopy (SEM) system. Utilizing Fourier transform infrared spectroscopy (FTIR), the functional groups on the nano-MgO were investigated. The Fourier transform infrared spectra of nanomaterials were recorded in the 400–4000 cm^−1^ wavelength range using KBr discs with TENSOR 27–Bruker (Billerica, MA, USA). The mineral composition of the nano-MgO was ascertained using X-ray diffraction (XRD), and the diffraction peaks were observed between 2θ = 15^◦^ and 2θ = 75^◦^. The specific surface area (SSA) was calculated by the Sauter formula: S = 6000/ρ × D, where S is the specific surface area, ρ is the density of the synthesized material, and D is the size of the particles.

### 2.10. Statistical Analysis

Analysis of variance (ANOVA) was performed on all data obtained from the pot experiment using PROC GLM of SAS 9.4 [33]. 

## 3. Results

### 3.1. Spectroscopic Analysis of MgO-NPs

The morphology of MgO NPs was demonstrated using TEM and SEM (Figure 3 and Figure 4). The MgO NPs were found to be spherical and rod-shaped, with some nanoparticle aggregation and diameters ranging from 26.35 to 32.69 nm, with an average of 29.47 nm. This was confirmed by the observed surface area of MgO NPs = (254.5 m^2^ g ^−^^1^). FTIR spectroscopy (Figure 5) provided information about the functional groups found in the MgO NPs sample. Peaks at 1448 cm^−^^1^, 881 cm^−^^1^, and 433 cm^−^^1^ were identified as Mg-O bonds, whereas a peak at 3700 cm^−^^1^ showed the presence of an O-H bond. The band at 461 cm^−^^1^ corresponds to the Si–O–Si bending vibration. The peaks at 2853 cm^−^^1^ and 2922 cm^−^^1^ are assigned to the O-H bond due to water molecules resulting from the calcination process. Furthermore, the peak at 3858 cm^−^^1^ is assigned to the asymmetric stretching vibration of –OH groups from Mg(OH)_2_. In the MgO-NPs sample, crystalline phases were found by XRD analysis (Figure 6). Magnesium planes are represented by significant diffraction peaks at 2θ values of 27.32°, 32.63°, 46.74°, 56.81°, 76.82°, and 85.12°. Peaks associated with magnesium compounds, such as magnesite (CMgO_3_), magnesium silicate (MgO_3_Si), magnesium potassium fluoride (F_3_KMg), and MgO_6_Pb_2_W, were also detected, indicating possible sites for arsenic adsorption on the MgO-NPs. X-ray diffraction analysis showed that silicon is present in the form of magnesium silicate and there may be traces of impurities with the material prepared from it, magnesium chloride. Also, FTIR analysis showed a peak at 1025 cm^−^^1^.

### 3.2. Maize Plant Biomass

Applying various isolate bacteria to sandy soil under arsenic stress, either with or without nano-MgO treatments, resulted in a considerable increase in maize plant biomass (Table 2). The total chlorophyll content increased from 8.90 μmol m^−^^2^ (control treatment) to 18.75, 19.95, 16.9, 17.57, 23.45, and 24.20 μmol m^−^^2^ for A, AB_1_, AB_2_, AN, AB_1_N, AB_2_N, and AB_1_B_2_ treatments, respectively. The highest total chlorophyll content was observed in sandy soil after applying the AB_1_B_2_ treatment. The treatments in this study were effective in improving total chlorophyll content in the following order: AB_1_B_2_ > AB_2_N > AB_2_ > AB_1_ > AB_1_N > AN > control.

The root length of the maize plant increased from 1.97 cm for the control treatment to 2.1, 2.1, 2.57, 3.17, 3.0, and 6.2 cm for A, AB_1_, AB_2_, AN, AB_1_N, AB_2_N, and AB_1_B_2_, respectively. The largest value of the root length of the maize plant was obtained in the AB_1_B_2_ application (Table 2). In this study, the root length of maize was not significantly different between AB_1_ and AB_2_.

The shoot length of the maize plant rose dramatically after the application of different isolate bacteria with or without nano-MgO amendments (Table 2). The shoot length of the maize plant increased by 17.8%, 6.7%, 11.1%, 11.0%, 13.3%, and 22.2% for A, AB_1_, AB_2_, AN, AB_1_N, AB_2_N, and AB_1_B_2_, respectively, in comparison to the control treatment. Among the treatments, the addition of AB_1_B_2_ resulted in the greatest and most notable increase in the shoot length of the maize plant. 

In pots treated with the addition of AB_1_, AB_2_, AN, AB_1_N, AB_2_N, and AB_1_B_2_ treatments, the fresh weight of the maize plants rose significantly (Table 2). The fresh weight of the maize plant rose from 6.3 (A) to 7.61 g plant^−^^1^ for AB_1_B_2_ and AN. There was no significant difference in the fresh weight of the maize plant between the AB_2_, AN, AB_1_N, AB_2_N, and AB_1_B_2_ treatments.

The addition of different isolate bacteria, either with or without nano-MgO additions, resulted in a significant increase in the total dry weight of the maize plant (Table 2). Compared to the treatment in the control, the dry weight of the maize plant increased by 17.6%, 18.6%, 20.95%, 20.95%, 21.9%, and 22.4% for A, AB_1_, AB_2_, AN, AB_1_N, AB_2_N, and AB_1_B_2_, respectively. When compared to other treatments, the addition of AB_1_B_2_ resulted in the largest and most significant increase in the dry weight of the maize plant. The study’s treatments were shown to be effective in improving the dry weight of the maize plant in the following order: AB_1_B_2_ > AB_2_N > AB_1_N = AN > AB_2_ > AB_1_> control.

### 3.3. Biochemical Characteristics

The data indicated that the application of B_1_, B_2_, and B_1_ + B_2_ strains reduced soil pH significantly while adding nano-MgO alone or with both PGPR strains (*R. nepotum* and *G. halophytocola*) increased soil pH (Table 3). The lowest pH was recorded with AB_1_B_2_, while the highest pH was recorded with the AN treatment. The mixture of two strains of PGPR bacteria had a synergistic effect on increasing soil organic matter (SOM) in the soil. The increase in organic matter results from increased root growth, which results in residues and secretions. Pots treated with the B_1_ + B_2_ strain mixtures achieved a 57.6% increase in SOM, greater than untreated ones. Soil organic matter for AB_2_ was higher than AB_1_ and AB_1_N by 1.19 and 0.95 times, respectively.

When the isolated bacteria from the study were applied, with or without nano-MgO additions, the microbial biomass carbon (MBC), total number of bacteria (CFU), and dehydrogenase activity (DHA) in sandy soil with a high degree of arsenic contamination increased significantly (Table 3). MBC levels in the soil showed a substantial difference among the treatments, ranging from 0.08 mg kg^−^^1^ in the control to 0.87 mg kg^−^^1^ in AB_1_B_2_ (Table 3). In contrast to the control treatment, the inclusion of AB_1_, AB_2_, AN, AB_1_N, AB_2_N, and AB_1_B_2_ increased MBC by 1.5, 4.5, 1.25, 2.38, 4.13, and 10.88 times, respectively. In this investigation, the MBC did not differ significantly between AB_2_ and AB_2_N. The MBC, CFU, and DHA content in pots treated with bacteria isolate B_2_ was higher than that in pots treated with bacteria isolate B_1_, whether added individually or with nano-MgO. Compared to the control treatment (Table 3), the addition of AB_1_, AB_2_, AN, AB_1_N, AB_2_N, and AB_1_B_2_ increased CFU by 1.45,1.63, 1.28, 1.49, 1.75, and 1.81 times, respectively. When compared to other treatments, the addition of AB_1_B_2_ showed the highest and most notable rise in MBC, CFU, and DHA content. The study’s treatments were shown to be effective in improving the MBC, CFU, and DHA in the following order: AB_1_B_2_> AB_2_N > AB_2_> AB_1_N > AB_1_ > AN > control.

### 3.4. Arsenic Available in Soil

In the sandy soil compared to As-stressed soil, the addition of AB_1_, AB_2_, AB_1_N, AB_2_N, and AB_1_B_2_ increased the available As by 5.1%, 10.1%, 7.2%, 12.9%, and 20.2%, respectively. However, the MgO-NPs treatment decreased the available As by 27.32% (Table 4). The study treatments showed their effectiveness in increasing available arsenic in sandy soil under As stress in the following order: AB_1_B_2_ > AB_2_N > AB_2_ > AB_1_N > AB_1_ > control. In this investigation, the content of available As did not differ significantly between AB_2_, AB_2_N, and AB_1_N.

### 3.5. Arsenic in Maize Plants

As shown in Table 4, the As content and uptake of maize plants declined noticeably in pots that were treated after the addition of the examined two strains of As-resistant PGPR with or without MgO-NPs. Compared to the therapy under control, the As content of the maize plant decreased by 18.64%, 23.83%, 33.80%, 31.21%, 36.62%, and 29.35% for AB_1_, AB_2_, AN, AB_1_N, AB_2_N, and AB_1_B_2_, respectively. The addition of MgO-NPs and B_2_ therapy led to the biggest and most notable reduction in the amount of As in the maize plant when compared to other treatments. The lowest As content and uptake of maize plants were found in AB_2_N-treated soils, with decreases of 36.62% and 22.76% relative to the control, respectively. The study’s treatments were shown to be effective in decreasing the As content and uptake of maize plants in the following order: AB_2_N > AN > AB_1_N > AB_1_B_2_ > AB_2_ > AB_1_ > control.

## 4. Discussion

The current study’s findings demonstrated that, by lowering the detrimental impacts of arsenic stress, the addition of MgO-NPs to severely arsenic-contaminated soil enhanced maize plant growth and biomass. The addition of MgO-NPs alone significantly improved the plant growth of maize compared to the control. According to Faizan et al. [34], applying MgO-NPs in the presence of As improved plant height and dry weight by 17% and 15%, respectively, and enhanced photosynthesis by 14.7%. This outcome coincides with their findings. Faizan et al. [34] discovered that the foliar treatment of MgO-NPs in As-stressed soybeans increased antioxidant enzymes while decreasing MDA and H_2_O_2_, resulting in enhanced cell membrane stability in the presence of MgO-NPs. Several antioxidant enzymes found in plant organelles help to detoxify reactive oxygen species (ROS). Our observations show that adding MgO-NPs to As-stressed maize enhanced the amount of chlorophyll (Table 2), as Mg is the key component in the tetrapyrrole loop of the chlorophyll molecule. Greater amounts of chlorophyll crops can capture and absorb more light energy because of their larger antennas [34,35]. Similarly, previous research [34] found that chlorophyll content increased after the foliar spraying of soybean plants with MgO-NPs.

According to [36], the addition of MgO-NPs improved plant growth, biomass, and chlorophyll content while reducing ROS levels, boosting the activity of antioxidant enzymes and reducing As accumulation in rice crops. According to Ahmed et al. [36], MgO-NPs have a unique effect on As translocation in rice plants due to their small size and large surface areas. Their application activates an antioxidative defense mechanism and reduces ROS accumulation by preventing its translocation from the soil to the rice plant. Wu et al. [37] observed similar findings: ZnO-NPs enhanced biomass, germination, and Zn uptake while reducing As uptake in rice. Similarly, Bidi et al. [38] found that nFe improved Fe uptake and strengthened the defenses against antioxidants in rice, indicating their major significance in reducing As phytotoxicity in plants. Furthermore, MgO-NP supplementation is an important strategy to mitigate arsenic stress in plants, as it reduces bioavailability and uptake in plants. Our extended X-ray absorption fine structure (EXAFS) investigation reveals that As(III) has been absorbed onto MgO, generating a layer of inner As atoms bonded to three oxygens as H_2_AsO_3_^−^, with additional As_2_O_3_ multilayer structures that are poorly adsorbed on the outside layer. As (V) adsorption typically involves HAsO_4_^2−^ coordination with four oxygen atoms [39]. The binding of magnesium oxide nanoparticles with arsenic is due to hydrogen bonding with the help of a negative charge between hydrogen atoms on (MgOH)_2_ and O atoms in arsenate. Also, the attraction of Mg^2+^ ions in the magnesium oxide nanoparticles as electrostatic to the negatively charged arsenate ions occurs [40].

*Rhizobim nepotum* and *G. halophytocola* increased the dry weight of maize plants by 17.6% and 18.6%, respectively. Similar results have been obtained by Chi et al. [41], who noticed that the total dry weight of rice (*Oryza sativa* L.) increased by almost 20% following inoculation with Rhizobium bacteria. Additionally, the traits of plant growth improved. Co-inoculation of *R. nepotum* and *G. halophytocola* strains enhanced fresh and dry weight output in maize plants in comparison to inoculation with *R. nepotum* or *G. halophytocola* alone. PGPR is important when plants require adaptation to a new environment, both naturally and in managed ecosystems [42]. The hormonal substances and helpful chemicals generated by these bacteria can help plants endure higher metal amounts. Thus, the plants would be well-suited to heavy metal-contaminated soil [43]. Zhang et al. [44] found that these microorganisms may enhance the biomass of plants in both direct and indirect ways. The direct process can convert components such as phosphorus and nitrogen into nutrients that are easily absorbed by plants. Indirectly, it can generate hormones and auxins such as cytokinin (CTK), indole-3-acetic acid (IAA), siderphores, and gibberellin (GA), which can directly increase root growth by encouraging plant cell elongation or accelerating cell division, thereby boosting plant biomass [45]. Rhizobium has shown remarkable PGP capabilities via a variety of plant growth promotion mechanisms, including phosphate solubilization, the fixation of nitrogen, and the production of siderophores, phytohormones, and exopolysaccharides. In addition, rhizobacteria emit VOCs, which are carbon-containing chemical compounds with low molecular masses [15]. Microbial volatiles have been shown to improve disease resistance and plant growth. Furthermore, they suppress soil-borne illnesses and strengthen the plant’s natural defense. *Glutamicibacter halophytocola* reduced the As content and uptake of maize plants by 23.83% and 9.82%, respectively. According to Mohsin et al. [46], microorganisms exhibit two unique processes for reducing arsenic: (i) external arsenic reduction in the periplasmic space and (ii) intracellular arsenic reduction that takes place in the cytoplasm. Microorganisms create exopolysaccharides (EPS) as defense biomacromolecules in the face of strong stress conditions, including drought, heat stress, lack of nutrients, and tolerance to salt [15,47]. Exopolysaccharides generated by *G. halophytocola* efficiently reduced NaCl stress in tomato plants by initiating a complicated regulatory system. Arati [47] showed that isolates (ASR-5, ASR-9, and ASR-14) can be used as bioremediation agents to lower the rates of As and improve the development aspects of tomato cultivars in As-contaminated sites under both in vitro and in vivo conditions.

In general, biosorption occurs when negatively charged EPSs interact with ions with positive charges (As, Cu, Pb, and Cd), making them immobile [48,49,50]. EPSs may play an important function in eliminating heavy metals from their surroundings as flocculants, as well as in adsorbing metal ions [51,52]. ATR-FTIR studies revealed that the functional groups of EPS (O-H, C=O, C-O, C=C, and N-O) were involved in the adsorption process of arsenic cations, with greater interactions among EPS and the arsenate than with arsenite [53]. The combination of the two strains (*R. nepotum* and *G. halophytocola*) showed a major response in all plant biomass variables, recording 24.2 μmol m^−2^, 6.2 cm, 18.33 cm, 7.61 g plant^−^^1^, and 2.57g plant^−^^1^ for total chlorophyll content, root length, shoot length, plant fresh weight, and dry weight, respectively. These findings are comparable to those reported by Li et al. [54]. The *G. halophytocola* strain (B_2_) or *R. nepotum* strain (B_1_), along with MgO-NPs, improves maize plant growth and decreases As uptake under arsenic stress. Although *G. halophytocola* or *R. nepotum* alone could increase chlorophyll content, decrease As content, and increase the uptake of maize plants, they had a synergistic effect when combined with MgO-NPs. A similar synergistic influence can be confirmed by decreasing H_2_O_2_ and arsenic uptake during arsenic stress, each of which is expected to alleviate stress. Rhizospheric bacteria modulate arsenic speciation, restrict translocation via root to shoot, and improve citric acid generation by roots, which in turn regulates soil pH and the bioavailability of arsenic in the soil. The interaction between the isolation of As-tolerant bacteria and nano-MgO with arsenic in soil is shown in Figure 7.

In this study, nano-bioremediation for arsenic treatment has many advantages, including low cost, environmental friendliness, wide application potential, and decreased clean-up time for contaminated areas. However, to avoid negative effects on the ecosystem and soil fauna and residual effects, detailed studies are needed.

## 5. Conclusions

MgO-NPs have a large surface area along with minerals and functional groups. They can also decrease As availability and As uptake and increase soil MBC and DHA and the total number of bacteria, all of which significantly enhance maize performance. The addition of AB_1_B_2_ gave the best chlorophyll content, SOM, MBC, DHA, and total number of bacteria compared to other treatments, which played an important role in increasing maize growth. The addition of *R. nepotum* and *G. halophytocola* alone or in combination with nano-MgO significantly decreased As uptake and increased the biological activity and development characteristics of maize plants cultivated in extremely arsenic-contaminated soil. The results recommend the use of *G. halophytocola* strain integration with nano-MgO to improve soil biochemical properties, reduce arsenic uptake, and increase maize growth.

## Figures and Tables

**Figure 1 nanomaterials-14-01164-f001:**
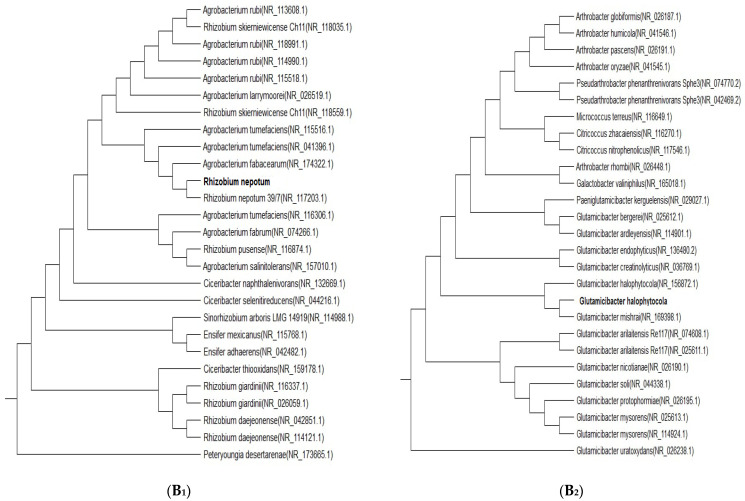
Neighbor-joining phylogenetic tree reconstructed on the basis of a 1.5 kp^b^ 16 S rRNA gene sequence showing the phylogenetic position of *Rhizobim nepotum* (**B_1_**) and *Glutamicibacter halophytocola* (**B_2_**).

**Figure 2 nanomaterials-14-01164-f002:**
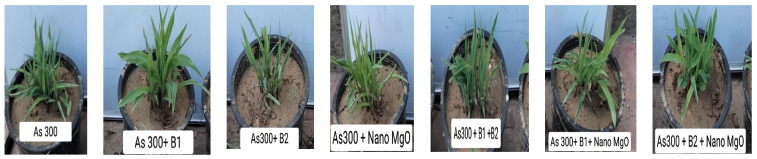
Maize growth during experimental period.

**Figure 3 nanomaterials-14-01164-f003:**
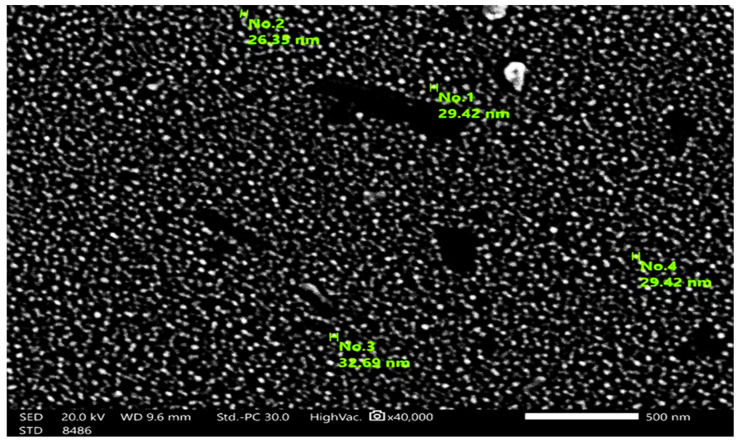
Transmission electron microscopy (TEM) of synthesized MgO-NPs.

**Figure 4 nanomaterials-14-01164-f004:**
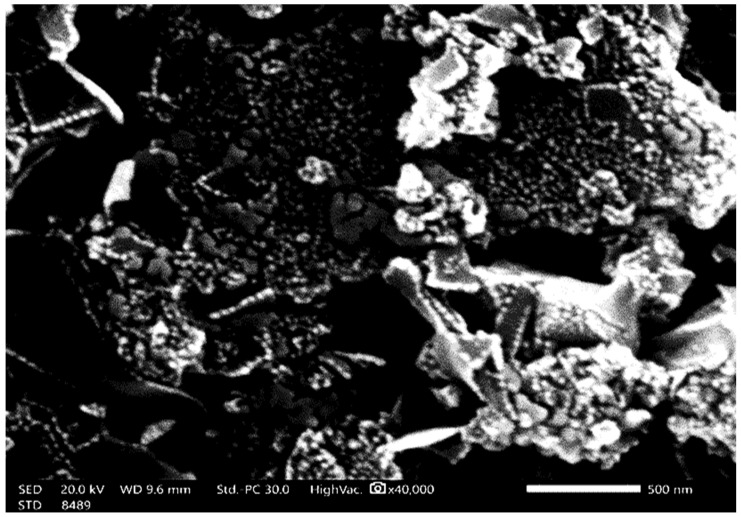
Scanning electron microscopy (SEM) of synthesized MgO-NPs.

**Figure 5 nanomaterials-14-01164-f005:**
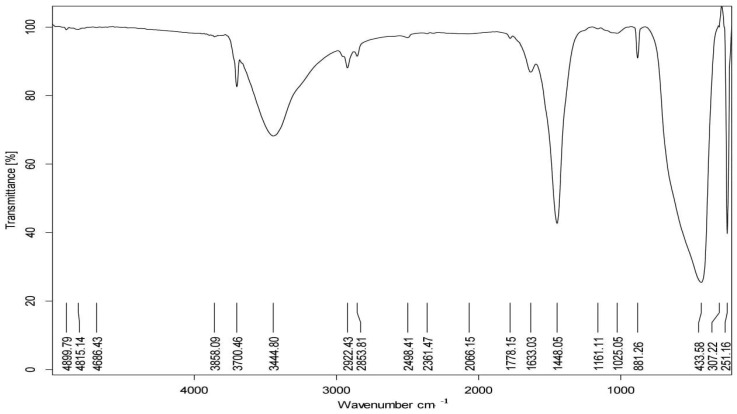
Fourier transform infrared spectroscopy (FTIR) of MgO-NPs.

**Figure 6 nanomaterials-14-01164-f006:**
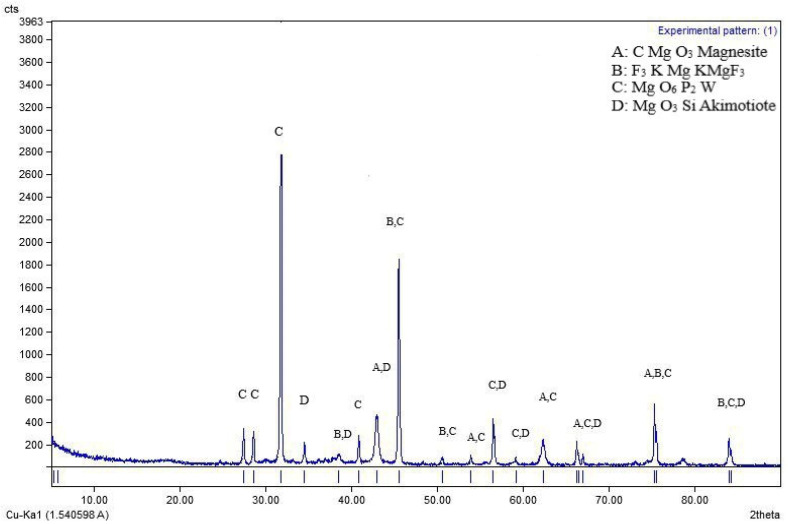
X-ray diffraction (XRD) of MgO-NPs.

**Figure 7 nanomaterials-14-01164-f007:**
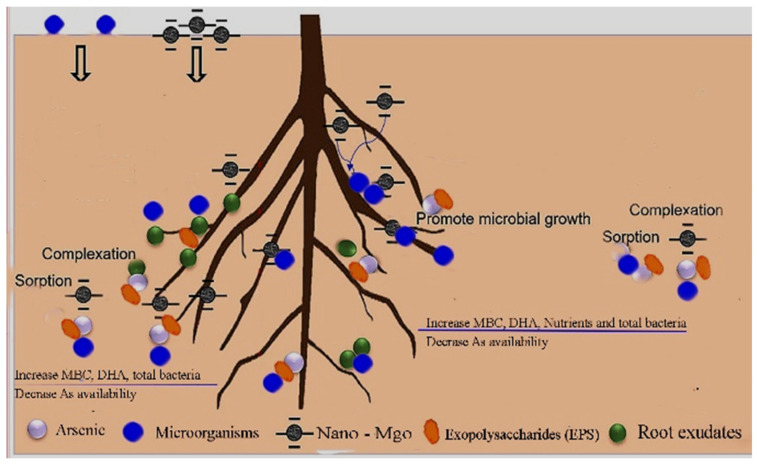
Interaction between isolation of Arsenic-tolerant bacteria and nano-MgO with arsenic in soil.

**Table 1 nanomaterials-14-01164-t001:** Physical and chemical properties of the soil before cultivation.

Chemical Characteristics	Unit	Value	Physical Characteristics	Unit	Value
EC	dS m^−1^	0.225	Particle size distribution		
pH (Soil suspension 1:2.5)		7.95	Clay	%	1.6
Soluble ions			Silt	%	1.7
Na^+^	mmol L^−1^	0.42	Sand	%	96.70
K^+^	mmol L^−1^	0.24	Texture class	Sandy	
Ca^+2^	mmol L^−1^	1.00	Field capacity	%	11.6
Mg^+2^	mmol L^−1^	0.57	Bulk density	g cm^−3^	1.48
Cl^−^	mmol L^−1^	1.20			
HCO_3_^−^	mmol L^−1^	0.92			
SO_4_^−2^	mmol L^−1^	0.11			
SOM ^a^	%	0.003			
MBC ^b^	%	0.002			
CaCO_3_	%	4.07			
CEC	cmol kg^−1^	4.65			
Total N		0.02			
Total P		0.07			
Total K	%	0.0002			
Arsenic (As)	mg kg^−1^	0.05			

SOM ^a^ Soil organic matter, MBC ^b^ Microbial biomass carbon, cation exchange capacity (CEC).

**Table 2 nanomaterials-14-01164-t002:** The integration effects of As (300 mg Kg^−^^1^), *R. nepotum* and *G. halophytocola* strains, and Nano-MgO on maize plant biomass after harvesting.

Treatments	Total Chlorophyll μmol m^−2^	Root Length cm	Shoot Length cm	Fresh Weight g plant^−1^	Dry Weight g plant^−1^
A	8.90 ± 1.25 ^g^	1.97 ± 0.25 ^e^	15.00 ± 2.65 ^f^	6.30 ± 0.06 ^c^	2.10 ± 0.02 ^b^
AB_1_	18.75 ± 1.25 ^d^	2.10 ± 0.15 ^e^	17.67 ± 2.89 ^b^	7.41 ± 0.06 ^b^	2.47 ± 0.02 ^a^
AB_2_	19.95 ± 0.15 ^c^	2.10 ± 0.10 ^e^	16.00 ± 2.65 ^e^	7.47 ± 0.01 ^ab^	2.49 ± 0.00 ^a^
AN	16.90 ± 0.80 ^f^	2.57 ± 0.31 ^d^	16.67 ± 0.58 ^d^	7.61 ± 0.02 ^a^	2.54 ± 0.01 ^a^
AB_1_N	17.57 ± 2.46 ^e^	3.17 ± 0.38 ^b^	16.67 ± 0.58 ^d^	7.37 ± 0.6 ^a^	2.54 ± 0.03 ^a^
AB_2_N	23.45 ± 1.05 ^b^	3.00 ± 0.00 ^c^	17.00 ± 3.00 ^c^	7.41 ± 0.03 ^b^	2.56 ± 0.02 ^a^
AB_1_ B_2_	24.20 ± 1.10 ^a^	6.20 ± 0.20 ^a^	18.33 ± 1.53 ^a^	7.61 ± 0.08 ^a^	2.57 ± 0.01 ^a^
LSD	2.46	0.43	3.46	1.62	0.32

Note: Column values with the same letters are statistical similar according to Duncan Multiple Range Test (DMRT) at *p* < 0.05, LSD: least significant difference.

**Table 3 nanomaterials-14-01164-t003:** The integration effects of As (300 mg Kg_−1_), R. nepotum and G. halophytocola strains, and Nano-MgO on biochemical soil characteristics after maize harvesting.

Treatments	pH	SOM %	MBC mg Kg^−1^	DHA mg TPF g^−1^ Soil day^−1^	Total Count of Bacteria CFU, log 10 g^−1^
A	7.51 ± 0.06 ^b^	0.59 ± 0.03 ^d^	0.08 ± 0.02 ^d^	61.67 ± 3.51 ^g^	3.51 ± 0.30 ^e^
AB_1_	7.34 ± 0.080 ^c^	0.76 ± 0.04 ^c^	0.12 ± 0.06 ^d^	139.67 ± 1.53 ^e^	5.08 ± 0.16 ^c^
AB_2_	7.02 ± 0.02 ^d^	0.91 ± 0.03 ^a^	0.36 ± 0.05 ^b^	182.33 ± 5.69 ^c^	5.73 ± 0.43 ^b^
AN	7.73 ± 0.10 ^a^	0.77 ± 0.03 ^c^	0.10 ± 0.04 ^d^	121.00 ± 2.65 ^f^	4.51 ± 0.34 ^d^
AB_1_N	7.69 ± 0.02 ^a^	0.86 ± 0.04 ^b^	0.19 ± 0.09 ^c^	148.33 ± 4.04 ^d^	5.26 ± 0.13 ^c^
AB_2_N	7.67 ± 0.01 ^a^	0.87 ± 0.01 ^a b^	0.33 ± 0.04 ^b^	189.00 ± 4.00 ^b^	6.15 ± 0.13 ^ab^
AB_1_ B_2_	6.81 ± 0.04 ^e^	0.92 ± 0.04 ^a^	0.87 ± 0.04 ^a^	206.33 ± 7.64 ^a^	6.35 ± 0.12 ^a^
LSD	0.07	0.05	0.09	5.07	0.44

Note: Column values with the same letters are statistical similar according to Duncan Multiple Range Test (DMRT) at *p* < 0.05, LSD: least significant difference.

**Table 4 nanomaterials-14-01164-t004:** Maize plant contents of As metal and its uptake as affected by As (300 mg Kg^−^^1^), *R. nepotum* and *G. halophytocola* strains, nano-MgO, and their combinations on maize plants biomass after harvesting.

Treatments	As Available mg Kg^−1^	As (mg/kg plant)	%	As Uptake (mg pots^−1^)	%
A	46.33 ± 0.58 ^f^	17.75 ± 0.58 ^e^		0.224 ^e^	
AB_1_	48.67 ± 0.58 ^d^	14.44 ± 0.58 ^c^	18.64	0.214 ^d^	4.46
AB_2_	51.00 ± 1.00 ^c^	13.52 ± 1.00 ^b^	23.83	0.202 ^c^	9.82
AN	33.67 ± 0.58 ^g^	11.75 ± 0.58 ^a^	33.80	0.18 ^a^	19.64
AB_1_N	49.67 ± 0.58 ^cd^	12.21 ± 0.58 ^ab^	31.21	0.186 ^b^	16.96
AB_2_N	52.33 ± 0.58 ^bc^	11.25 ± 1.00 ^a^	36.62	0.173 ^a^	22.76
AB_1_B_2_	55.67 ± 0.58 ^a^	12.54 ± 0.58 ^ab^	29.35	0.193 ^bc^	13.83
LSD	1.14	1.26		0.01	

Note: Column values with the same letters are statistical similar according to Duncan Multiple Range Test (DMRT) at *p* < 0.05, LSD: least significant difference.

## Data Availability

The datasets used and analyzed during the current study are available from the corresponding author upon reasonable request.

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
