# Peer review of "Nano-Bioremediation of Arsenic and Its Effect on the Biological Activity and Growth of Maize Plants Grown in Highly Arsenic-Contaminated Soil"

_nanomaterials, 2024, doi:10.3390/nano14131164_

Round 1
Reviewer 1 Report
Comments and Suggestions for Authors
The paper shows R. nepotum and G. halophy-tocola can improving the biological activity and growth parameters of maize plants. The mechanisms about improving the biological activity and growth parameters of maize plants were discussed by several technologies. The studies were quite systematic and the resulted were well organized by the authors. I’d like to recommend the publication of this nanomaterials after revision.
1. According to morphology results, the mapping results should be provided to identify element distribution.
2. For morphology, SEM and TEM image should be change to high resolution.
3. For FTIR results, peaks at 2853 cm-1, 2922 cm-1, and 3858 cm-1 should be explain what they are and their source.
4. The mechanism of G. halophytocola or R. nepotum in plant should be described in detail.
Author Response
Reviewer #1: Reviewer 1 comments
Thanks very much for your efforts and useful comments about our manuscript.
Comments and Suggestions for Authors
The paper shows R. nepotum and G. halophy-tocola can improving the biological activity and growth parameters of maize plants. The mechanisms about improving the biological activity and growth parameters of maize plants were discussed by several technologies. The studies were quite systematic and the resulted were well organized by the authors. I’d like to recommend the publication of this nanomaterials after revision.
- According to morphology results, the mapping results should be provided to identify element distribution.
Elemental distribution map (EDS) analysis of MgO-NPs using energy-dispersive spectroscopy (EDS) was not available.
- For morphology, SEM and TEM image should be change to high resolution.
We improved with high resolution
- For FTIR results, peaks at 2853 cm-1, 2922 cm-1, and 3858 cm-1 should be explain what they are and their source.
We added (The peaks at 2853 cm-1, 2922 cm-1 is assigned to O-H bond due to water molecules resulting from the calcination process. And also, the peak at 3858 cm-1 is assigned to the asymmetric stretching vibration of –OH groups from Mg(OH)2).
- The mechanism of G. halophytocola or R. nepotum in plant should be described in detail.
We added (Rhizobium has showed remarkable PGP capabilities via a variety of plant growth promotion mechanisms, including phosphate solubilization and the production of siderophores, phytohormones, and exopolysaccharides. In addition, rhizobacteria emit VOCs, which are carbon-containing chemical compounds with low molecular masses [15]. Microbial volatiles have been shown to improve disease resistance and plant growth).
Thank you

Reviewer 2 Report
Comments and Suggestions for Authors
Review report
Dear authors, I read your article entitled -Nano-bioremediation of arsenic and its effect on the biological activity and growth of maize plants grown in highly arsenic- contaminated soil.
My observations are the following:
1. The title should be re-worded so as to highlight the novelty of the study and attract the reader.
2. In the introduction part to the general objective of the study highlight the novelty and originality of the study should be added. In addition to what you have written about what you present (i.e.: The goals of this study is to examine the impact of As-tolerant bacterial isolates via. bacterial R. nepotum strain (B1), bacterial G. halophytocola strain (B2), nano-MgO (N) and their combinations on the As content, biological activity, and growth parameters of maize plants grown in highly As-contaminated soil (300 mg As Kg-1 ). Is necessary to mention if it is for the first time the study presented in the literature what is new compared to the literature, etc. ,
4. Explain schematically the retention of As in the soil based on the system developed.
5. Discuss the advantages and disadvantages of using these systems in soil remediation.
6. Put the literature data in a table at the experimental part is lost the importance of the presented study.
7. Do you have pictures from the experiment to put in the article (about the plant, etc)? Please add such pictures.
8. Please increase the resolution of the images so that they are clearer.
9. The English needs to be improved and some of the editorial mistakes need to be corrected.
Author Response
Reviewer #2: comments
Thanks very much for your efforts and useful comments about our manuscript.
Dear authors, I read your article entitled -Nano-bioremediation of arsenic and its effect on the biological activity and growth of maize plants grown in highly arsenic- contaminated soil.
My observations are the following:
- The title should be re-worded so as to highlight the novelty of the study and attract the reader.
Some authors in the research register the title before submission it, and any change causes a problem for them.
- In the introduction part to the general objective of the study highlight the novelty and originality of the study should be added. In addition to what you have written about what you present (i.e.: The goals of this study is to examine the impact of As-tolerant bacterial isolates via. bacterial R. nepotum strain (B1), bacterial G. halophytocola strain (B2), nano-MgO (N) and their combinations on the As content, biological activity, and growth parameters of maize plants grown in highly As-contaminated soil (300 mg As Kg-1). Is necessary to mention if it is for the first time the study presented in the literature what is new compared to the literature, etc. ,
We added in introduction (Therefore, the novelty of this research is the use of Rhizobim nepotum, Glutamicibacter halophytocola, MgO-NPs, and their combinations in highly arsenic-contaminated soil to reduce arsenic content and increase maize growth. The present study assumed that the addition of MgO-NPs alone would not be able to completely reduce arsenic content in highly arsenic-contaminated soil and increase maize growth. Therefore, the aim of the study was to investigate the impact of As- tolerant bacterial isolates via. bacterial R. nepotum strain (B1), bacterial G. halophytocola strain (B2), nano-MgO (N) and their combinations on the As content, biological activity, and growth parameters of maize plants grown in highly As-contaminated soil (300 mg As Kg-1)).
And also added in the middle introduction (Traditional soil remediation strategies for As such as removal excavation and landfilling) and containment (capping) are either high cost of removal and loss of land use, the use of nanotechnology and biological-based techniques for on-site arsenic removal is desirable due to cost and sustainability issues that have not yet been studied together)
- Explain schematically the retention of arsenic in the soil based on Nano-bioremediation
The interaction between Isolation of Arsenic-Tolarant Bacteria and nano-MgO with arsenic in soil is very complex and needs further study
- Discuss the advantages and disadvantages of using these systems in soil remediation.
We added at the end of discussion (In this study, Nano-bioremediation for arsenic treatment has many advantages, including low cost, environmental friendliness, wide application potential, and shortening the clean-up time for contaminated areas. However, to avoid negative effects on the ecosystem, soil fauna, and residual effects, detailed studies are needed).
- Put the literature data in a table at the experimental part is lost the importance of the presented study.
We moved the table
- Do you have pictures from the experiment to put in the article (about the plant, etc)? Please add such pictures.
We added
- Please increase the resolution of the images so that they are clearer.
We improved
- The English needs to be improved and some of the editorial mistakes need to be corrected.
All manuscript has been thoroughly reviewed and improved by Dr. Nermin Ibrahim; Email: nermeen.ibrahim@art.menofia.edu.eg; Ph.D in Applied Linguistics. Lecturer in Linguistics; Department of English language and literature, Faculty of Arts, and Menofia University, Egypt
Thank you

Reviewer 3 Report
Comments and Suggestions for Authors
The manuscript relates to the “Nano-bioremediation of arsenic and its effect on the biological activity and growth of maize plants grown in highly arsenic-contaminated soil”.
Language revision is not necessary.
Please, see and consider the following comments and suggestions for amendments and additions to the manuscript. All answers should be included in the revised text.
1. “Abstract and the whole text: Please, use SOM as abbreviation for soil organic matter.
2. Ls.56-9: Not clear.
3. Ls.60-1: Please, explain: strategies and byproducts.
4. Ls.61-4: Not clear.
5. Ls.67-8: the criterion for categorizing polysaccharides is not uniform. Internal polysaccharides?
6. Ls.75-7: Language revision needed.
7. Ls.83-4: Not clear.
8. L.98: Reactivity capacity?
9. L.102: nZVI?
10. L.104: cefdinir?
11. Table 1: Are there any other heavy metals contained?
12. “Bacteria Isolation and Purification” Section: Method not clear. Add the exact method and quantities.
13. Ls. 147-8: explain the washing.
14. L.155: Define the As concentrations.
15. Ls.156-7: Define the exact quantities.
16. L.157: Why use distilled water? Plant take useful metal ions from water. Tap water does not contain As.
17. L.158: Explain “field capacity”.
18. L.166: Explain “DTPA”.
19. Ls. 180-1: Not clear; explain the electronic scale.
20. L.205: How was the surface area calculated?
21. Ls.208-9: Explain the origin of Si. Why not see the characteristic peak at about 1000 cm-1? Is it actually Si present?
22. Ls.218-21: Where did all these constituents come from?
23. Ls.218-21: Why are they considered as possible adsorption sites? Why is adsorption onto these materials favored?
24. L.233: maize spinach plant?
25. L.240-1: Please, correct.
26. L.264: Explain why and how SOM increases.
27. Section “Arsenic in soil and maize plants”: Explain As availability and how it was calculated.
28. Section “Arsenic in soil and maize plants”: Are the contents of these two paragraphs consistent with each other? Explain.
29. Table 4: Define “As Uptake (mg pots-1) and how it was calculated.
30. Table 4: Are these As contents accepted when edible products are encountered?
31. Ls.372-7: Not compatible; arsenate and arsenite anions (AsO33-and AsO43-) are negatively charged; besides does As exist as As3+?
Comments on the Quality of English LanguageModerate editing of English language required
Author Response
Reviewer #3: comments
Thanks very much for your efforts and useful comments about our manuscript.
Comments and Suggestions for Authors
The manuscript relates to the “Nano-bioremediation of arsenic and its effect on the biological activity and growth of maize plants grown in highly arsenic-contaminated soil”.
Language revision is not necessary.
Please, see and consider the following comments and suggestions for amendments and additions to the manuscript. All answers should be included in the revised text.
- “Abstract and the whole text: Please, use SOM as abbreviation for soil organic matter.
We corrected
- 56-9: Not clear.
We improved
- 60-1: Please, explain: strategies and byproducts and Ls.61-4: Not clear.
We rewritten again
- 67-8: the criterion for categorizing polysaccharides is not uniform. Internal polysaccharides?
Polysaccharides, including internal polysaccharides, functional polysaccharides, and extracellular or exopolysaccharides (EPS).
- 75-7: Language revision needed.
We revision
- 83-4: Not clear.
We improved
- 98: Reactivity capacity?
We corrected to high adsorption capacity
- 102: nZVI?
We improved to Němeček et al. [21] found that the combination of nanoscale zero-valent iron (nZVI) and whey reduces Cr(VI) to about 97–99%.
- 104: cefdinir?
We added a semisynthetic cephalosporin antibiotic
- Table 1: Are there any other heavy metals contained?
Only arsenic
- “Bacteria Isolation and Purification” Section: Method not clear.
We improved
- 147-8: explain the washing.
Five times with distilled water
- 155: Define the As concentrations.
300 mg L-1 of arsenic from sodium arsenic (AsHNa2O4.7H2O)
- 156-7: Define the exact quantities.
We corrected
- 157: Why use distilled water? Plant take useful metal ions from water. Tap water does not contain As.
Top water in this place is groundwater which contains a percentage of arsenic.
- 158: Explain “field capacity”.
Field capacity is the amount of soil moisture or water content held in the soil after excess water has drained off and the rate of downward movement has decreased significantly, which usually occurs within 2-3 days after irrigation.
- 166: Explain “DTPA”.
We added diethylenetriaminepentaacetic acid
- 180-1: Not clear; explain the electronic scale.
We changed to Using Digital Weighing Scales
- 205: How was the surface area calculated?
Specific surface area (SSA) was calculated by the Sauter formula: S = 6000/ρ×D
Where S is the specific surface area, ρ is the density of the synthesized material, and D is the size of the particles
- Ls.208-9: Explain the origin of Si. Why not see the characteristic peak at about 1000 cm-1? Is it actually Si present? 17. Ls.218-21: Where did all these constituents come from?
X-ray diffraction analysis showed that silicon is present in the form of magnesium silicate and there may be traces of impurities with the material prepared from it, magnesium chloride. Also, FTIR analysis showed a peak at 1025 cm-1
- 218-21: Why are they considered as possible adsorption sites? Why is adsorption onto these materials favored?
We added (The binding of magnesium oxide nanoparticles with arsenic is due to hydrogen bonding with the help of negative charge between hydrogen atoms on (MgOH)2 and O atoms in arsenate. Also, the attraction of Mg2+ ions in the magnesium oxide nanoparticles as electrostatic to the negatively charged arsenate ions occurs [40]).
- 233: maize spinach plant?
We corrected
- 240-1: Please, correct.
We corrected.
- 264: Explain why and how SOM increases.
The increase in organic matter results from increased root growth, which results in residues and secretions.
- Section “Arsenic in soil and maize plants”: Explain As availability and how it was calculated.
We added in Materials and Methods (Available arsenic was extracted using diethylenetriaminepentaacetic acid (DTPA) solution and measured using an inductively coupled plasma atomic emission spectrometer (ICP-ISO Prodigy Plus) with calibration by standard curve. A stock solution of arsenic at a concentration of 1.000 g L-1 was used to prepare the standards. The calibration was adjusted in the range of 0-10 μg L-1
- Section “Arsenic in soil and maize plants”: Are the contents of these two paragraphs consistent with each other? Explain.
We separated them from each other
- Table 4: Define “As Uptake (mg pots-1) and how it was calculated.
The plant As uptake (influx into plants) (mg As pot−1) was calculated as dry weight of plant and As concentration of plants (total As content).
As uptake (mg As pot−1) = (DW*%As/100)*1000
- Table 4: Are these As contents accepted when edible products are encountered?
The World Health Organization has determined that arsenic concentrations in plants grown in soils without arsenic range from 0.02 to 5 mg/kg (dry weight). The daily intake of total arsenic from foods and beverages generally ranges in the range of 20–300 μg/day.
- Ls.372-7: Not compatible; arsenate and arsenite anions (AsO33-and AsO43-) are negatively charged; besides does As exist as As3+?
We added Extended X-ray absorption fine structure (EXAFS) investigation reveals that As(III) has been absorbed onto MgO, generating a layer of inner As atoms bonded to three oxygens as H2AsO3−, with additional As2O3 multilayer structures that are poorly adsorbed on an outside layer. The binding of magnesium oxide nanoparticles with arsenic is due to hydrogen bonding with the help of negative charge between hydrogen atoms on (MgOH)2 and O atoms in arsenate. Also, the attraction of Mg2+ ions in the magnesium oxide nanoparticles as electrostatic to the negatively charged arsenate ions occurs [40]
Thank you

Reviewer 4 Report
Comments and Suggestions for Authors
Review Report on
Nano-bioremediation of arsenic and its effect on the biological activity and growth of maize plants grown in highly arsenic-contaminated soil
I read carefully the whole manuscript, it looks suitable for the publication after careful examination. However, often found the typo and grammatical errors and flaws in the manuscript. If those mistake will be rectified by the authors then it will pass for the publication.
Here are my Queries.
· Table 1, what is ECe, it should be clarified.
· Is it HCO3- or HCO3-?
· Molecular identification of two isolates:
· Using the 16S rRNA gene sequence (1.5 kpb) to identify two isolates of bacteria that are most tolerant to high concentrations of arsenic, up to 300 mg L-1, which are Rhizobim nepo-tum (B1) and Glutamicibacter halophytocola (B2) (Fig.1). (authors should provide suitable references, since this is important for the this research work)n
· Preparation of Nano-MgO, the protocol is not satisfactory, when NaOH is dissolved in 50 mL of ethanol, its an exothermic reaction and forms immediately NaOEt? Then added MgCl2 to NaOEt while stirring? Not clear? Check it.
· Fig 4 shown, several functional groups however, in the write up, the authors have determined very few peaks? What are others?
· (FTIR spectroscopy (Fig. 4) gave information about the functional groups found in the MgO NPs sample. Peaks at 1448 cm-1, 881 cm-1, and 433 cm-1 were identified as Mg-O bonds, whereas a peak at 3700 cm-1 showed the presence of O-H bond. The band at 461 cm-1 is corresponds to the Si–O–Si bending vibration. )
Result: Minor Rivision
Comments on the Quality of English Language
Review Report on
Nano-bioremediation of arsenic and its effect on the biological activity and growth of maize plants grown in highly arsenic-contaminated soil
I read carefully the whole manuscript, it looks suitable for the publication after careful examination. However, often found the typo and grammatical errors and flaws in the manuscript. If those mistake will be rectified by the authors then it will pass for the publication.
Here are my Queries.
· Table 1, what is ECe, it should be clarified.
· Is it HCO3- or HCO3-?
· Molecular identification of two isolates:
· Using the 16S rRNA gene sequence (1.5 kpb) to identify two isolates of bacteria that are most tolerant to high concentrations of arsenic, up to 300 mg L-1, which are Rhizobim nepo-tum (B1) and Glutamicibacter halophytocola (B2) (Fig.1). (authors should provide suitable references, since this is important for the this research work)n
· Preparation of Nano-MgO, the protocol is not satisfactory, when NaOH is dissolved in 50 mL of ethanol, its an exothermic reaction and forms immediately NaOEt? Then added MgCl2 to NaOEt while stirring? Not clear? Check it.
· Fig 4 shown, several functional groups however, in the write up, the authors have determined very few peaks? What are others?
· (FTIR spectroscopy (Fig. 4) gave information about the functional groups found in the MgO NPs sample. Peaks at 1448 cm-1, 881 cm-1, and 433 cm-1 were identified as Mg-O bonds, whereas a peak at 3700 cm-1 showed the presence of O-H bond. The band at 461 cm-1 is corresponds to the Si–O–Si bending vibration. )
Result: Minor Rivision
Author Response
Reviewer #4: comments
Thanks very much for your efforts and useful comments about our manuscript.
Comments and Suggestions for Authors
Nano-bioremediation of arsenic and its effect on the biological activity and growth of maize plants grown in highly arsenic-contaminated soil
I read carefully the whole manuscript, it looks suitable for the publication after careful examination. However, often found the typo and grammatical errors and flaws in the manuscript. If those mistake will be rectified by the authors then it will pass for the publication.
Here are my Queries.
Table 1, what is ECe, it should be clarified.
We changed to EC
Is it HCO3- or HCO3-?
Ok HCO3-
Molecular identification of two isolates:
- 1. Using the 16S rRNA gene sequence (1.5 kpb) to identify two isolates of bacteria that are most tolerant to high concentrations of arsenic, up to 300 mg L-1, which are Rhizobim nepo-tum (B1) and Glutamicibacter halophytocola (B2) (Fig.1). (authors should provide suitable references, since this is important for the this research work)n
Genomic DNA of the test bacterial isolates grown on nutrient broth was extracted with Gene Jet Bacterial Genomic DNA Extraction Kit (Fermentas). The 16 S rRNA gene of the isolate was amplified by the polymerase chain reaction (PCR) using universal primers at Sigma Scientific Services Co., Giza, Egypt.
- 2. Preparation of Nano-MgO, the protocol is not satisfactory, when NaOH is dissolved in 50 mL of ethanol, its an exothermic reaction and forms immediately NaOEt? Then added MgCl2 to NaOEt while stirring? Not clear? Check it.
We rewritten again as (Two solutions were prepared, the first by dissolving 20 mM of MgCl2 in 50 ml of ethanol. The second is by dissolving 40 mM of NaOH in 50 ml of ethanol. Both solutions were stirred at 60 °C for 30 min. During stirring for 10 minutes, the second solution was added dropwise to the first solution at 60 °C, diluting the solution by adding 20 ml of ethanol with continuous stirring. Then the previous solution was centrifuged at 2000 rpm for 10 min, and the resulting residue was dried at 60 °C in a hot air oven. Then the resulting compound was calcined at 300 °C for 1 hour to obtain crystalline MgO nanoparticles).
- 3. Fig 4 shown, several functional groups however, in the write up, the authors have determined very few peaks? What are others?
We added (The peaks at 2853 cm-1, 2922 cm-1 is assigned to O-H bond due to water molecules resulting from the calcination process. And also, the peak at 3858 cm-1 is assigned to the asymmetric stretching vibration of –OH groups from Mg(OH)2).
Thank you

Round 2
Reviewer 2 Report
Comments and Suggestions for Authors
Dear authors, I have read the answers to my questions at Nano-bioremediation of arsenic and its effect on the biological activity and growth of maize plants grown in highly arsenic contaminated soil.
Please complete the review report with all the answers added in manuscript (for example at Question 6. Do you have pictures from the experiment to put in the article? Please add such pictures. Response 6: We added (where is the figure added in rapport?) and at the same time please answer at questions no.1 and 3 (Can't explain the interactions based on the literature? You don't need in-depth studies).
At question 5. Put the literature data in a table at the experimental part is lost the importance
of the presented study.
Response 5. We moved the table. Why do you move the literature data? Why does the data not appear in the table?
Author Response
Reviewer #2: comments
Thanks very much for your efforts and useful comments about our manuscript.
response of all commentsin attached file as pdf

Reviewer 3 Report
Comments and Suggestions for Authors
The manuscript relates to the “Nano-bioremediation of arsenic and its effect on the biological activity and growth of maize plants grown in highly arsenic-contaminated soil”.
The following answers should be included in the revised text.
- 102: nZVI?
We improved to Němeček et al. [21] found that the combination of nanoscale zero-valent iron (nZVI) and whey reduces Cr(VI) to about 97–99%.
- 104: cefdinir?
We added a semisynthetic cephalosporin antibiotic
- Table 1: Are there any other heavy metals contained?
Only arsenic
- 147-8: explain the washing.
Five times with distilled water
- 155: Define the As concentrations.
300 mg L-1 of arsenic from sodium arsenic (AsHNa2O4.7H2O)
- 157: Why use distilled water? Plant take useful metal ions from water. Tap water does not contain As.
Top water in this place is groundwater which contains a percentage of arsenic.
- 158: Explain “field capacity”.
Field capacity is the amount of soil moisture or water content held in the soil after excess water has drained off and the rate of downward movement has decreased significantly, which usually occurs within 2-3 days after irrigation.
- 205: How was the surface area calculated?
Specific surface area (SSA) was calculated by the Sauter formula: S = 6000/ρ×D
Where S is the specific surface area, ρ is the density of the synthesized material, and D is the size of the particles
- Ls.208-9: Explain the origin of Si. Why not see the characteristic peak at about 1000 cm-1? Is it actually Si present? 17. Ls.218-21: Where did all these constituents come from?
X-ray diffraction analysis showed that silicon is present in the form of magnesium silicate and there may be traces of impurities with the material prepared from it, magnesium chloride. Also, FTIR analysis showed a peak at 1025 cm-1
- 264: Explain why and how SOM increases.
The increase in organic matter results from increased root growth, which results in residues and secretions.
- Table 4: Define “As Uptake (mg pots-1) and how it was calculated.
The plant As uptake (influx into plants) (mg As pot−1) was calculated as dry weight of plant and As concentration of plants (total As content).
As uptake (mg As pot−1) = (DW*%As/100)*1000
- Table 4: Are these As contents accepted when edible products are encountered?
The World Health Organization has determined that arsenic concentrations in plants grown in soils without arsenic range from 0.02 to 5 mg/kg (dry weight). The daily intake of total arsenic from foods and beverages generally ranges in the range of 20–300 μg/day
Author Response
Thanks very much for your efforts and useful comments about our manuscript.
Comments and Suggestions for Authors
The manuscript relates to the “Nano-bioremediation of arsenic and its effect on the biological activity and growth of maize plants grown in highly arsenic-contaminated soil”.
The following answers should be included in the revised text.
- 102: nZVI?
We improved to Němeček et al. [21] found that the combination of nanoscale zero-valent iron (nZVI) and whey reduces Cr(VI) to about 97–99%.
We included in page 3 line 98 to 99
- 104: cefdinir?
We added a semisynthetic cephalosporin antibiotic
We included in page 3 line 102
- Table 1: Are there any other heavy metals contained?
Only arsenic
We included in page 4 in table 1
- 147-8: explain the washing.
Five times with distilled water
We included in page 5 line 164
- 155: Define the As concentrations.
300 mg L-1 of arsenic from sodium arsenic (AsHNa2O4.7H2O)
We included in page 6 line 171
- Why use distilled water? Plant take useful metal ions from water. Tap water does not contain As.
We added because top water in this place is groundwater which contains a percentage of arsenic.
We included in page 6 line 174 and 175
- 158: Explain “field capacity”.
Field capacity is the amount of soil moisture or water content held in the soil after excess water has drained off and the rate of downward movement has decreased significantly, which usually occurs within 2-3 days after irrigation.
We included in page 6 line 176 to 178
- 205: How was the surface area calculated?
Specific surface area (SSA) was calculated by the Sauter formula: S = 6000/ρ×D
Where S is the specific surface area, ρ is the density of the synthesized material, and D is the size of the particles
We included in page 7 line 221 to 224
- Ls.208-9: Explain the origin of Si. Why not see the characteristic peak at about 1000 cm-1? Is it actually Si present? 17. Ls.218-21: Where did all these constituents come from?
X-ray diffraction analysis showed that silicon is present in the form of magnesium silicate and there may be traces of impurities with the material prepared from it, magnesium chloride. Also, FTIR analysis showed a peak at 1025 cm-1
we included in page 7 line 245 to 247
- Explain why and how SOM increases.
The increase in organic matter results from increased root growth, which results in residues and secretions.
We included in page 11 line 305 and 306
18.Table 4: Define “As Uptake (mg pots-1) and how it was calculated.
The plant As uptake (influx into plants) (mg As pot−1) was calculated as dry weight of plant and As concentration of plants (total As content).
As uptake (mg As pot−1) = (DW*%As/100)*1000
We included in page 7 line 207 to 210
- Table 4: Are these As contents accepted when edible products are encountered?
The World Health Organization has determined that arsenic concentrations in plants grown in soils without arsenic range from 0.02 to 5 mg/kg (dry weight). The daily intake of total arsenic from foods and beverages generally ranges in the range of 20–300 μg/day
We included in page 2 line 54 to 57
Thank you

Round 3
Reviewer 2 Report
Comments and Suggestions for Authors
The authors answered most of my questions. In my opinion, the article can be accepted for publication.
For the future, I ask the authors to complete the review report with the data added to the article.
Reviewer 3 Report
Comments and Suggestions for Authors
No comments